# Psychopathological Impact in Patients with History of Rheumatic Fever with or without Sydenham’s Chorea: A Multicenter Prospective Study

**DOI:** 10.3390/ijerph191710586

**Published:** 2022-08-25

**Authors:** Alessandro Orsini, Thomas Foiadelli, Attilio Sica, Andrea Santangelo, Niccolò Carli, Alice Bonuccelli, Rita Consolini, Sofia D’Elios, Nicolò Loddo, Alberto Verrotti, Giuseppe Di Cara, Chiara Marra, Maria Califano, Anna Fetta, Marianna Fabi, Stefania Bergamoni, Aglaia Vignoli, Roberta Battini, Marta Mosca, Chiara Baldini, Nadia Assanta, Pietro Marchese, Gabriele Simonini, Edoardo Marrani, Francesca Felicia Operto, Grazia Maria Giovanna Pastorino, Salvatore Savasta, Giuseppe Santangelo, Virginia Pedrinelli, Gabriele Massimetti, Liliana Dell’Osso, Diego Peroni, Duccio Maria Cordelli, Martina Corsi, Claudia Carmassi

**Affiliations:** 1Pediatric Neurology, Pediatric University Department, Azienda Ospedaliera Universitaria Pisana, University of Pisa, 56121 Pisa, Italy; 2Pediatric Clinic, Fondazione IRCCS Policlinico San Matteo, University of Pavia, 27100 Pavia, Italy; 3Department of Pediatrics, University of Perugia, 06123 Perugia, Italy; 4Child Neurology Unit, University of Bologna, 40126 Bologna, Italy; 5Childhood and Adolescence Neurology and Psychiatry Unit, ASST GOM Niguarda, 20121 Milan, Italy; 6Health Sciences Department, Università degli Studi di Milano, 20121 Milan, Italy; 7Department of Clinical and Experimental Medicine, University of Pisa, 56121 Pisa, Italy; 8Department of Developmental Neuroscience, Istituto di Ricerca e Cura a Carattere Scientifico (IRCCS) Fondazione Stella Maris, 56121 Pisa, Italy; 9Heart Hospital, G. Monasterio Tuscan Foundation, 54100 Massa, Italy; 10Pediatric Rheumatology, Meyer Children Hospital, University of Florence, 50134 Florence, Italy; 11Child and Adolescent Neuropsychiatry Unit, Department of Medicine, Surgery and Dentistry, University of Salerno, 84084 Baronissi, Italy; 12Pediatric Clinic, ASST di Crema, 26013 Crema, Italy; 13Child Neuropsychiatry Unit, ISMEP—P.O. Cristina—ARNAS Civico, Via dei Benedettini 1, 90100 Palermo, Italy; 14Occupational Health Department, Azienda Ospedaliero Universitaria Pisana, Via Paradisa 2, 56124 Pisa, Italy

**Keywords:** Sydenham’s chorea, acute rheumatic fever, neuropsychiatric tests, persistent and recurrent chorea

## Abstract

Sydenham’s chorea (SC) is a post-streptococcal autoimmune disorder of the central nervous system, and it is a major criterium for the diagnosis of acute rheumatic fever (ARF). SC typically improves in 12–15 weeks, but patients can be affected for years by persistence and recurrencies of both neurological and neuropsychiatric symptoms. We enrolled 48 patients with a previous diagnosis of ARF, with or without SC, in a national multicenter prospective study, to evaluate the presence of neuropsychiatric symptoms several years after SC’s onset. Our population was divided in a SC group (n = 21), consisting of patients who had SC, and a nSC group (n = 27), consisting of patients who had ARF without SC. Both groups were evaluated by the administration of 8 different neuropsychiatric tests. The Work and Social Adjustment Scale (WSAS) showed significantly (*p* = 0.021) higher alterations in the SC group than in the nSC group. Furthermore, 60.4% (n = 29) of the overall population experienced neuropsychiatric symptoms other than choreic movements at diagnosis and this finding was significantly more common (*p* = 0.00) in SC patients (95.2%) than in nSC patients (33.3%). The other neuropsychiatric tests also produced significant results, indicating that SC can exert a strong psychopathological impact on patients even years after its onset.

## 1. Introduction

Sydenham’s chorea (SC) is a post-streptococcal autoimmune disorder of the central nervous system. It has a widely acknowledged autoimmune pathogenesis and it is one of the major clinical criteria for the diagnosis of acute rheumatic fever (ARF), according to Jones Criteria [1,2,3].

The choreic movements after which the pathology is named are brief, rapid, irregular, jerking, and non-stereotypic movements of the limbs, often associated to restlessness of the trunk. In addition to them, SC also features several other neurologic symptoms, as hypotonia, migraine, dysarthria, dysgraphia, dysphagia, and lingual dystonia. Despite mainly being known for its neurological manifestations, SC is also characterized by many neuropsychiatric symptoms (as also happens for many other SNC pathologies [4,5]), the most common ones being emotional lability, anxiety, attention deficit hyperactivity disorder, mood disorders, and obsessive-compulsive symptoms [6,7,8].

As a result of these heterogeneous manifestations, SC diagnosis can be difficult, especially in an emergency setting [9,10]. Clinical findings are important to direct the diagnosis, together with other manifestations of ARF and evidence of recent GAS infection, even though the presence of the latter is not mandatory. Diagnosis confirmation can be achieved thanks to laboratory tests, in particular inflammatory markers (e.g., CRP, ESR), and an elevation of antistreptolysin-O (ASLO) and anti-DNase B titers. Electroencephalographic and neuroimaging abnormalities are common, but nonspecific [11,12,13]. However, they can be useful for excluding differential diagnosis [14,15,16,17].

Available treatments can be divided into three groups: antibiotics, symptomatic drugs, and immunomodulatory drugs [15,18,19]. Antibiotic therapy consists of penicillin G benzathine, preferably administered in its long-acting intramuscular formulation every 21 to 28 days. Symptomatic and immunomodulatory drugs might not be necessary at all in mild SC cases, while for severe disease there is a wide choice of therapeutic alternatives: symptomatic treatments are divided into dopamine antagonists (e.g., haloperidol, pimozide, and risperidone) and antiepileptic drugs (e.g., valproic acid and carbamazepine), while immunosuppressive treatments are mainly represented by corticosteroids [20,21], but also include intravenous immunoglobulin [22] and plasma exchange [23], exactly as they are used in other neurological disease [24]. Despite this wide range of therapeutic choices, there are still no standardized therapies or official guidelines for SC treatment [25,26].

With regards to SC prognosis, it typically improves gradually, with symptoms lasting from 12 to 15 weeks [27]. Although almost all patients achieve complete remission, with or without therapy, in several cases recurrence and persistence of symptoms have been reported even years after the initial bout [28,29,30]. In particular, neuropsychiatric symptoms are mainly reported in the acute phase of the disease, but there are a few studies that show them to precede or follow the acute disease onset, even by years [14,31].

To further investigate this topic, we conducted a national multicenter prospective study evaluating the presence of neuropsychiatric symptoms in patients with a previous diagnosis of rheumatic fever, with or without Sydenham’s chorea. We studied a population of 48 patients several years after the onset of symptoms, to better clarify even if there are differences between ARF with SC and without SC in terms of acute and chronic neuropsychiatric symptoms.

## 2. Materials and Methods

This is a national multicentric prospective study which involved Pediatric Neurology Units, Pediatric Rheumatology Units or Pediatric Neuropsychiatric Units of 7 different hospitals located all over Italy.

Recruitment was conducted between September 2021 and March 2022. We recruited patients between the ages of 16 and 40 years old who received a diagnosis of Rheumatic Fever (diagnosed according to the revised Jones Criteria) with or without Sydenham’s chorea at least 2 years before the enrollment date. Exclusion criteria were inadequate knowledge of Italian language, other verbal communication limitations that could prevent patients from completing the evaluations, and insufficient collaboration skills.

Social and demographic variables (date of birth, gender, level of education and employment status) were collected for all participants. We also asked the recruiting centers to send us information about comorbidities (rheumatic cardiac involvement and other diseases), ongoing therapies, prior psychological counseling, age at symptoms onset, neurological and neuropsychiatric symptoms at diagnosis (obsessive-compulsive disorder, mood or anxiety disorders, psychotic disorders, tics, hyperactivity, behavior disorders), instrumental examinations (neurophysiological and neuroradiological examinations), laboratory measurements (TASLO, anti-DNase antibodies), received treatments (antiepileptic drugs, antipsychotic drugs, corticosteroids, IVIG, plasmapheresis), recovery time (duration of both the neurological and psychiatric symptoms), and follow-up (administration of neuropsychological tests, onset of movement and autoimmune disorders).

Assessments included 8 neuropsychiatric measures: Patient Health Questionnaire-9 (PHQ-9) and DSM-5 Level 2-Depression were used to evaluate depression symptoms; anxiety symptoms were assessed by the administration of Generalized Anxiety Disorder 7-Item (GAD-7), and DSM-5 Level 2-Anxiety; Trauma and Loss Spectrum-Self Report (TALS-SR) was used to investigate the presence of PTSD manifestations; alterations of work and social functioning were evaluated by the administration of the Work and Social Adjustment Scale (WSAS); Structured Clinical Interview for the DSM-5-Clinical Version (SCID-V-CV) and DSM-5 Self-Rated Level 1 Cross-Cutting Symptom Measure were both used to investigate multiple neuropsychiatric pathologies. All measures were administered in Italian, using the officially translated versions of the tools.

Data analysis was aimed at correlating the presence of a previous diagnosis of Sydenham’s chorea with clinical manifestations, with the execution of diagnostic and therapeutic procedures, with follow-up findings and the neuropsychiatric tests scores. Descriptive statistics were used for a first screening of the investigated variables. A chi-square test (or Fisher test when appropriate) was used to compare categorical variables as socio-demographical and clinical characteristics and neuropsychological tests results between individuals with and without Sydenham’s chorea. For the quantitative comparisons of the non-parametric independent samples, the Mann-Whitney U test was used. In particular, Mann-Whitney U test was used to analyze the scores obtained by SC and nSC patients when administered some of the neuropsychiatric questionnaires and to compare the disappearance time of neurological and neuropsychiatric symptoms in SC and nSC patients. Furthermore, several models of decision trees and multiple logistic regression were executed to investigate the possible influence of sociodemographic, clinical and treatment-related factors on the incidence of neuropsychiatric symptoms alongside and in combination the presence of SC. Statistical analyses were carried out using SPSS software; *p* < 0.05 were considered statistically significant.

## 3. Results

### 3.1. Sample Background Characteristics

A total of 48 patients (25 males and 23 females), 27 with Sydenham’s chorea (SC) and 21 with rheumatic fever without Sydenham’s Chorea (nSC), were recruited into our study. The mean age of the recruited patients was of 21.6 ± 4.35 years, while the mean age at symptoms onset was of 10.36 ± 2.47 years.

In our cohort of patients, Sydenham’s chorea was significantly more frequent in female sex (*p* = 0.045): 60.9% (n = 14) of patients with Sydenham’s chorea were females, against only 39.1% (n = 9) of the nSC patients. Thus, female patients were more likely than male patients to have Sydenham’s chorea.

Among the 47 out of 48 patients who provided information about their occupational status, 68.1% (n = 32) were still studying at the time of enrollment, while 27.7% (n = 13) were working and 4.3% (n = 2) were unemployed.

With regards to comorbidities, 75% (n = 36) of patients had rheumatic cardiac involvement, while 27.1% (n = 13) of the total population was affected by other diseases. A chi-square test of independence was performed to examine the relation between the presence of chorea and cardiac involvement. The relation between these variables was significant, X2 (1, N = 48) = 4.769, *p* = 0.029, showing that SC patients were less likely than nSC patients to have cardiac involvement.

### 3.2. Clinical Data

Moreover, 60.4% (n = 29) of patients were reported to have experienced neuropsychiatric or neurologic symptoms other than choreic movements. This finding was significantly more common in SC patients (95.2%) than in nSC patients (33.3%), as showed by the execution of a chi-square test, X2 (1, N = 48) = 16.429, *p* < 0.001.

Focusing on neurologic symptoms other than chorea, they were observed in 29.2% (n = 14) of the total population and they were more common in SC patients (61.9%) than in nSC patients (3.7%). This result was shown to be statistically significant by the execution of a chi-square test of independence, X2 (1, N = 48) = 16.653, *p* < 0.001. Thus, SC patients were more likely than nSC patients to have neurologic symptoms other than chorea (Table 1).

On the other hand, neuropsychiatric symptoms were present in 52.1% (n = 25) of the overall population and they were more common in SC patients (76.2%) than in nSC patients (33.3%). The execution of a chi-square test of independence (X2 (1, N = 48) = 7.061, *p* = 0.008) showed that SC patients were significantly more likely than nSC patients to have neuropsychiatric symptoms (Table 2).

Taking a closer look at neuropsychiatric symptoms, mood or anxiety disorders were the most common ones, since they were reported in 41.7% (n = 20) of the overall population, while obsessive-compulsive disorder (OCD) was reported in 4.2% (n = 2), psychotic symptoms in 4.2% (n = 2), tics in 16.7% (n = 8), hyperactivity in 4.2% (n = 2) and behavior disorders in 12.5% (n = 6). None of these psychiatric symptoms, individually considered, were found to be significantly more common in SC patients than in nSC patients when analyzed by chi-square tests. However, statistical significance was almost reached by mood and anxiety symptoms, X2 (1, N = 48) = 2.634, *p* = 0.105.

Furthermore, 37.5% (n = 18) of the enrolled patients took sessions of psychological counselling before enrollment time. A chi-square test of independence was performed to examine the relation between the presence of chorea and the psychological counselling, but the relation between these variables was not significant, X2 (1, N = 48) = 2.489, *p* = 0.115.

With regards to the diagnostic process, 27.1% (n = 13) of patients were administered neuropsychological tests at diagnosis and this finding was significantly more common among SC patients (50.0%) than among nSC patients (11.1%), as showed by the execution of a chi-square test of independence, X2 (1, N = 47) = 6.849, *p* = 0.009. Moreover, neuropsychological tests were only used by 3 out of the 7 centers which took part to this study and only one of them administered them to nSC patients too.

Furthermore, we collected the values of TASLO and anti-DNase antibodies measured at diagnosis for 27 and 15 patients, respectively. The minimum TASLO value was 300 U/mL, while its maximum observed value was of 1880 U/mL, with a mean of 654.37 ± 454.15 U/mL. The minimum anti-DNase antibodies value was 10.0 U/mL, while its maximum observed value was of 3757.7 U/mL, with a mean of 547.03 ± 929.84 U/mL. It was not possible to demonstrate any statistically significant difference between SC and nSC patients neither for TASLO nor for anti-DNase antibodies values.

We also collected data about therapeutic choices in our population, and we compared our findings between SC and nSC patients. 64.6% (n = 31) of the patients received treatments other than antibiotics, and this finding was significantly (*p* = 0.017) more common in SC patients (85.7%) than in nSC ones (48.1%), as showed by the execution of a chi-square test of independence, X2 (1, N = 48) = 5.738, *p* = 0.017.

Among the 18 SC patients who received additional therapy to benzathine G penicillin, 6 (33.3%) were administered symptomatic treatment alone, 5 (27.8%) received immunomodulant treatment alone, and 7 (38.9%) received them both.

In particular, 10.4% (n = 5) of the overall population was treated with antiepileptic drugs, 22.9% (n = 11) was treated with antipsychotic drugs, 52.1% (n = 25) was treated with corticosteroids, 8.3% (n = 4) was treated with intravenous immunoglobulins (IVIG), and 4.2% (n = 2) was treated with plasmapheresis. Among these treatments, antiepileptic drugs (*p* = 0.012), antipsychotic drugs (*p* < 0.001), and IVIG (*p* = 0.031) were significantly more used in SC patients than in nSC patients, as showed by the execution of Fisher’s exact chi-square tests.

We also tried to understand how different treatments were used in different centers. Focusing on symptomatic therapies (antiepileptic drugs and antipsychotic drugs), they were of course administered to SC patients only, but we observed a certain variability among the 7 centers that took part to our study: 2 centers only used antipsychotic drugs, 2 centers only used antiepileptic drugs (although one of these only treated 1 out of 6 patients), 2 centers used both antiepileptic and antipsychotic drugs, and 1 center did not use symptomatic therapy at all.

With regards to immunomodulant treatment, the most used drugs were corticosteroids (52.1%), which were administered to both SC and nSC patients in similar proportions (57.1% of SC patients and 48.1% of nSC patients received this treatment). However, it is noteworthy that 2 out of the 7 centers which took part to this study did not use corticosteroids to treat any of the referred patients. Furthermore, 4 out of 7 centers used IVIG to treat one SC patient each, while the usage of plasmapheresis was only reported by one center which used it to treat one SC patient and one nSC patient.

Regarding the follow-up, 12.5% (n = 6) of the overall population was reported to have developed autoimmune diseases during or after the end of the follow-up period, while 16.7% (n = 8) was reported to have developed movement disorders in the same period. Neuropsychological tests were administered to 6.3% (n = 3) of 47 patients during the follow-up period, all of which were being treated at the same center. We also tried to investigate the disappearance time of neurological and neuropsychiatric symptoms, observing that, among the 22 patients who were reported experiencing neurological symptoms (chorea included), 36.4% (n = 8) of them had symptoms lasting less than one month, 36.4% (n = 8) were symptomatic for 1–6 months, 18.2% (n = 4) had symptoms lasting for more than six months, while 9.1% (n = 2) of them were reported to have ongoing neurological symptoms.

Among the 14 patients who had neuropsychiatric symptoms with a known disappearance time, 21.4% (n = 3) of them had symptoms lasting less than 1 month, 57.1% (n = 8) were symptomatic for 1–6 months, and 21.4% (n = 3) of them had symptoms for more than 6 months. It was not possible to demonstrate any significant difference in disappearance time of neurologic and neuropsychiatric symptoms between SC and nSC patients.

Lastly, in order to investigate the possible influence of sociodemographic, clinical and treatment-related factors on the incidence of neuropsychiatric symptoms alongside and in combination the presence of SC, we executed several models of decision trees and multiple logistic regression with the incidence of neuropsychiatric symptoms as the outcome, and sociodemographic, clinical, and treatment-related factors as variables of interest. The model featuring “Presence of SC”, “Gender”, “Comorbidities”, “Age of symptoms onset”, and “Treatment” as independent variables, and “Neuropsychiatric symptoms” as the dependent variable was found to be the best one, with an overall percentage of correct classification of 83%. In this model, the only significant variable was the presence of SC (*p* = 0.004), with a b coefficient of 3.63 (standard error = 1.28), corresponding to an odds ratio of 37.57 (Appendix A). However, the high number of independent variables compared to the sample size could be an important limitation of this analysis that must be taken into account.

### 3.3. Neuropsychiatric Tests

Data deriving from the analysis of the answers given by SC and nSC patients to the 8 administered neuropsychiatric measures will be shown in this paragraph.

Regarding the total WSAS score, SC patients (mean = 2.48, S.D. = 0.512, mean rank = 28.69) reported significantly more work and social functioning difficulties than nSC patients (mean = 2.19, S.D. = 0.483, mean rank = 21.24) at the Independent-Samples Mann-Whitney U Test, U(N0 = 27, N1 = 21) = 371.500, z = 2.313, *p* = 0.021 two-tailed (Figure 1) (Appendix A). WSAS is made up of 5 questions exploring 5 different domains of work and social functioning, so we decided to further investigate these domains to understand whether some of them are significantly altered in SC patients when compared to nSC patients. Thus, by the execution of Independent-Samples Mann-Whitney U Tests, we demonstrated that SC patients were significantly more likely to have a compromission of WSAS domains 3 (private leisure) and 5 (close relationships) (Appendix A).

In fact, WSAS question 3 (private leisure) scores of SC patients (mean = 1.94, S.D. = 1.514, mean rank = 23.67) were higher than those of nSC patients (mean = 0.90, S.D. = 1.410, mean rank = 15.75). A Mann-Whitney test indicated that this difference was statistically significant, U(N0 = 20, N1 = 18) = 255.000, z = 2.285, *p* = 0.028 two-tailed.SC patients (mean = 2.00, S.D. = 1.609, mean rank = 23.97) obtained higher scores than nSC patients (mean = 0.90, S.D. = 1.210, mean rank = 15.48) in WSAS question 5 (close relationships), too. The distributions in the two groups were shown to differ significantly by the execution of a Mann-Whitney test, U(N0 = 20, N1 = 18) = 260.500, z = 2.447, *p* = 0.017, two-tailed. No statistical significance was shown for WSAS questions 1 (*p* = 0.740), 2 (*p* = 0.158), and 4 (*p* = 0.133) (Figure 2) (Appendix A).

Only 54.2% (n = 26) of the overall population completed the TALS-SR. A chi-square test of independence was performed to examine the relation between the presence of chorea and PTSD diagnosis obtained at the TALS-SR test. The relation between these variables, investigated by Fisher’s Exact Test (because some cells have expected count less than 5) was not significant, *p* = 0.095.We also used Fisher’s Exact Test to investigate the presence of Partial PTSD in our population, but statistical significance was not reached neither for Partial A PTSD (*p* = 0.356), nor for Partial B PTSD (*p* = 1.000), nor for Partial PTSD in general (A or B) (*p* = 0.395). Fisher’s Exact Test was also used to examine the relation between the presence of chorea and a diagnosis of complete or partial (A or B) PTSD obtained at the TALS-SR test. The relation between these variables was not significant: *p* = 0.701.

SC patients reported similar levels of anxiety to nSC patients, according to both the GAD7 and the DSM-5 Level 2-Anxiety measures, analyzed by the Independent-Samples Mann-Whitney U Test. In particular, when administered the GAD-7, SC patients (mean = 1.86, S.D. = 1.014, mean rank = 27.05) did not report significantly different levels of anxiety compared to nSC ones (mean = 1.48, S.D. = 1.014, mean rank = 22.52), as shown by the execution of a Mann-Whitney test, U(N0 = 27, N1 = 21) = 337,000, z = 1163, *p* = 0.245 two-tailed. The same was found analyzing by the execution of a Mann-Whitney test the answers obtained by administering DSM-5 Level 2-Anxiety to SC (mean = 1.10, S.D. = 1.338, mean rank = 24.10) and nSC (mean = 1.10, S.D. = 1.338, mean rank = 23.92) patients, U(N0 = 26, N1 = 21) = 275,000, z = 0.048, *p* = 0.962 two-tailed. However, in the former case a stronger trend can be observed (Appendix A).

SC patients reported similar levels of depression to nSC patients, according to both the PHQ-9 and the DSM-5 Level 2-Depression measures, analyzed by the execution of Independent-Samples Mann-Whitney U Tests. In fact, there was not a significant difference between SC (mean = 0.95, S.D. = 0.865, mean rank = 26.10) and nSC (mean = 0.81, S.D. = 0.962, mean rank = 23.26) patients regarding the scores obtained at PHQ-9, U(N0 = 27, N1 = 21) = 317.000, z = 0.746, *p* = 0.456 two-tailed. This result was also confirmed by the answers of SC (mean = 0.57, S.D. = 1.076, mean rank = 23.86) and nSC (mean = 0.57, S.D. = 1.00, mean rank = 25.00) patients to the DSM-5 Level 1-Depression test, U(N0 = 27, N1 = 21) = 270.000, z = −0.361, *p* = 0.718 two-tailed (Appendix A).

We used the Independent-Samples Mann-Whitney U Test to independently compare the scores obtained by SC and nSC patients in the 13 domains of the DSM-5 Self-Rated Level 1 Cross-Cutting Symptom Measure. None of them reached statistical significance, but the mean ranks of SC patients were almost always higher than those of nSC patients, particularly in domain III (${\mathrm{SC}}_{meanrank}$ = 27.43; ${\mathrm{nSC}}_{meanrank}$ = 22.22; U(N0 = 27, N1 = 21) = 345.000, z = 1.335, *p* = 0.182 two-tailed), in domain IV (${\mathrm{SC}}_{meanrank}$ = 27.95; ${\mathrm{nSC}}_{meanrank}$ = 21.81; U(N0 = 27, N1 = 21) = 356.000, z = 1.559, *p* = 0.119 two-tailed), and in domain XII (${\mathrm{SC}}_{meanrank}$ = 27.38; ${\mathrm{nSC}}_{meanrank}$ = 22.26; U(N0 = 27, N1 = 21) = 344.000, z = 1.345, *p* = 0.179 two-tailed). These results suggest that statistical significance could be reached with a higher sample size. We also tried to dichotomize the results obtained for all the 13 independent domains, assigning a positive value to all the cases with a score ≥ 2 and a negative value to all the cases with a score < 2, in order to be able to analyze them by the execution of chi-square tests. However, no significant differences between SC and nSC patients were observed (Appendix A).

SCID-5 was only completed by 30 patients (17 SC and 13 nSC) out of 48. The heterogeneity of the collected results does not allow us to highlight any prominent diagnosis neither among the SC patients nor among the nSC patients (Appendix A).

The main investigated variables are summarized in Table 3.

## 4. Discussion

### 4.1. Sample Background Characteristics

The predominance of female gender observed in our population was in accordance with the literature [32], while the mean age at the time of SC presentation (10.36 ± 2.47 years) was slightly higher than the expected, since the peak incidence is generally considered 8 to 9 years [32] and a previous study on a large Italian cohort observed a mean age of 9.35 years [6]. Regarding occupational status and level of education, most of the patients in our overall population were still studying (68.1%) or working (27.7%), and only 2 patients (4.3%) were unemployed, with no significant differences between SC and nSC groups. Thus, rheumatic fever and Sydenham’s chorea do not seem to have affected unemployment rate in the long term in our patients, but further research will be needed to investigate possible influences on academic and work performance.

### 4.2. Clinical Data

The literature reports neurological manifestations in approximately two-thirds of patients with Sydenham’s chorea, the most common ones being dysgraphia and dysarthria [6]. Our study confirms this finding, since in the SC group neurologic symptoms were observed in 61.9% of patients. Moreover, this percentage was significantly higher than the one observed in the nSC group (3.7%).

With regards to neuropsychiatric symptoms, they were observed in 52.1% (n = 25) of the overall population. Focusing on the SC group, neuropsychiatric symptoms were found in 76.2% of patients. This percentage is rather higher than the one reported by previous studies (51%) on SC patients [6], and significantly higher than the one we observed in nSC patients (33.3%). However, when we further investigated neuropsychiatric symptoms individually, none of them proved to be significantly more common in SC patients than it was in nSC patients. In accordance with literature [6], in our study the most common neuropsychiatric symptoms were mood and anxiety disorders, which were reported by 41.7% of the overall population, with a noteworthy difference (*p* = 0.105) between SC (57.1%) and nSC patients (29.6%). For this reason, studies on more numerous samples could be useful in determining whether SC patients are more likely than nSC patients to develop any neuropsychiatric symptom in particular as well as we perform with other disease [4,24]. It should also be noted that obsessive-compulsive disorder was only reported by 1 SC patient (4.8%) and this data strongly differs from what was reported by some studies (70%) [33], even though it is in accordance with what was already reported in Italy by previous studies (3.1%) [6].

TASLO and anti-DNase titers at the first visit were collected for 27 and 15 patients, respectively, showing a great heterogeneity.

Regarding the treatment of Sydenham’s chorea, previous research reported a great heterogeneity throughout Italy, observing that, although almost all patients are treated with benzathine G penicillin every 21 days, there is a huge variability in the other therapeutic choices, with noteworthy differences between different centers [6]. Our study supports these statements. In fact, we observed that 18 out of the 21 enrolled SC patients (85.7%) received additional therapies to antibiotics, similarly to what had been reported by previous Italian studies [6], and significantly (*p* = 0.017) more than what we observed in nSC patients (48.1%). However, among these 18 SC patients, we observed very heterogeneous therapeutic choices, with 6 patients (33.3%) who received symptomatic treatment alone, 5 patients (27.8%) who were administered immunomodulant treatment alone, and 7 patients (38.9%) who received them both. Thus, our findings are consistent with the gradual shift towards immunomodulant treatment of Sydenham’s chorea which is taking place all over the world thanks to the growing evidence of its efficacy [6,34,35], as is also happening for many other SNC diseases as their immunopathogenesis is understood [24].

Regarding patients’ follow-up, we observed that only one of the seven centers administered neuropsychological tests to SC patients during the follow-up period. Moreover, it was not possible to demonstrate any significant difference in disappearance time of neurologic and neuropsychiatric symptoms between SC and nSC patients, which could also be partly due to the small number of patients with known disappearance time of symptoms. However, it is interesting to note that all the patients with ongoing symptoms or with symptoms lasting more than six months had Sydenham’s chorea. Lastly, we observed no differences among SC and nSC patients in the rate of patients that developed autoimmune disease or movement disorders during or after the end of the follow-up period.

### 4.3. Neuropsychiatric Tests

The analysis of the results deriving by the administration of the WSAS test to the overall study population allowed us to demonstrate a statistically significant impairment in work and social functioning among SC patients, when compared to nSC patients. Further investigating the 5 domains that constitute the WSAS test, we also showed that the enjoyment of private leisure and the ability to form and maintain close relationships (familiar relationships in particular) were significantly more impaired in SC patients than in nSC patients.

Two of the seven centers that took part to this study did not administer the TALS-SR to their patients, so the fact that it was completed only by about a half (n = 26) of the overall population could be the cause of its non-statistically significant results. We tried to overcome this problem by considering not only complete PTSD, but partial PTSD too [36,37]. However, our search for a correlation between partial PTSD and Sydenham’s chorea led to even fewer encouraging results, probably because the less restrictive criteria for partial PTSD diagnosis made its diagnosis common in nSC patients too.

Mood disorders and anxiety resulted to be the most common neuropsychiatric symptoms in SC patients at diagnosis time, but their persistence years after diagnosis does not seem to be correlated to Sydenham’s chorea, as shown by the fact that none of the 4 measures (GAD7 and DSM-5 Level 2-Anxiety for anxiety; PHQ9 and DSM-5 Level 2-Depression for depression) administered to our population produced scores that significantly differed between SC and nSC patients.

Moreover, the analysis of the scores obtained by the 13 independent domains of the DSM-5 Self-Rated Level 1 Cross-Cutting Symptom Measure did not show statistically significant differences between SC and nSC patients. The absence of statistical significance in the depression domain (I) and in the anxiety domain (IV) is in line with what we observed by administering the other measures, while the fact that the personality functioning domain (XII) was one of the most promising ones reflects the significant results obtained with the WSAS.

Lastly, only 5 out of the 7 centers that took part to this study administered SCID-5 to their patients, probably because it’s the only measure that must be administered verbally by trained personnel and its complete administration may take even longer than one hour. Thus, we collected SCID-5 results for 30 out of 48 patients, but the small sample size and the heterogeneity of the obtained results did not allow for the identification of significantly predominant disorders in SC patients when compared to nSC patients.

## 5. Conclusions

This is a national multicentric prospective study, whose aim is to evaluate the presence of psychiatric conditions in patients with a previous diagnosis of rheumatic fever, with or without Sydenham’s chorea, several years after its clinical onset.

Most of our results are in accordance with what has been observed by previous studies, and particularly by the ones conducted on Italian cohorts of patients [6,21]. Some noteworthy exceptions are the high mean age of our overall population, the high prevalence of neuropsychiatric symptoms in SC patients, and the low prevalence of obsessive-compulsive disorder.

We observed several statistically significant differences between SC and nSC patients, the most relevant ones being gender, rheumatic cardiac involvement, neurologic and neuropsychiatric symptoms at diagnosis, diagnostic and therapeutic choices, WSAS total scores and WSAS domains 3 (private leisure) and 5 (close relations) scores.

We also noticed several differences between the centers that took part to our study, underlining the already known heterogeneity in diagnosis, treatment, and follow-up of patients with Sydenham’s chorea.

Rheumatic fever and Sydenham’s chorea do not seem to have affected unemployment rate in the long term, but this observation needs to be confirmed by further research.

Many other promising data were collected, but the rarity of the investigated disease did not allow us to enroll a larger population of patients, which could have provided an adequate statistical power to obtain other significant results.

Major awareness should be raised among physicians, patients, and caregivers on SC neuropsychiatric manifestations. In fact, their early detection could be important in both younger patients in developmental age and in older patients, because their long-term symptoms could remain undiagnosed and their correlation with SC could be neglected. Thus, a more widespread knowledge on this topic could be important to guarantee appropriate psychotherapeutic and pharmacological treatments to all patients. Moreover, since pediatric chronic diseases have been shown to variably affect caregivers’ mental health [38], the possibility to offer SC children’s parents psychological support programs should be considered.

Thus, further studies with higher sample sizes will be needed to evaluate these observations.

## Figures and Tables

**Figure 1 ijerph-19-10586-f001:**
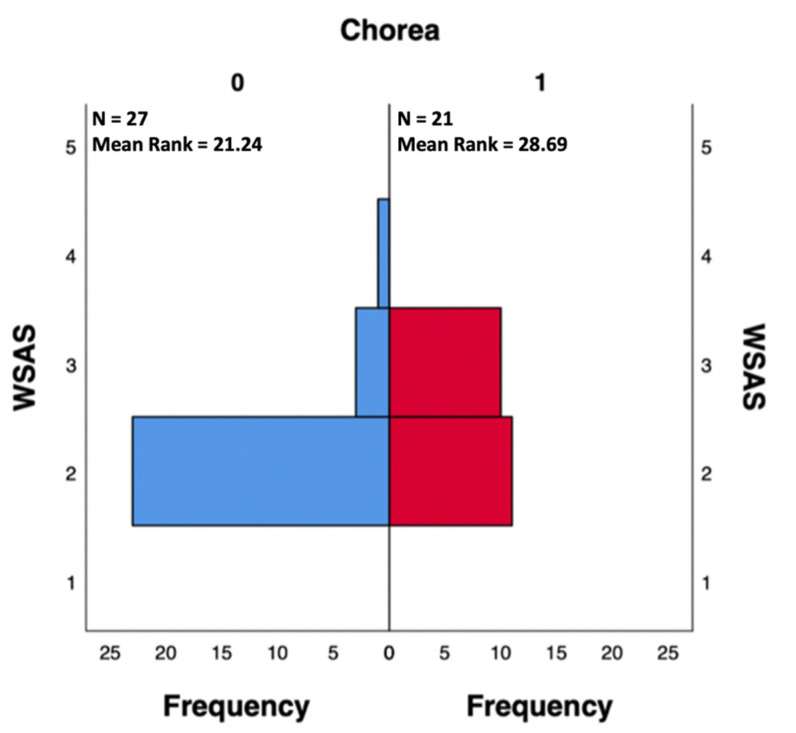
Independent-Samples Mann-Whitney U Test reporting significantly different WSAS scores between SC (Chorea 1) and nSC (Chorea 0) patients.

**Figure 2 ijerph-19-10586-f002:**
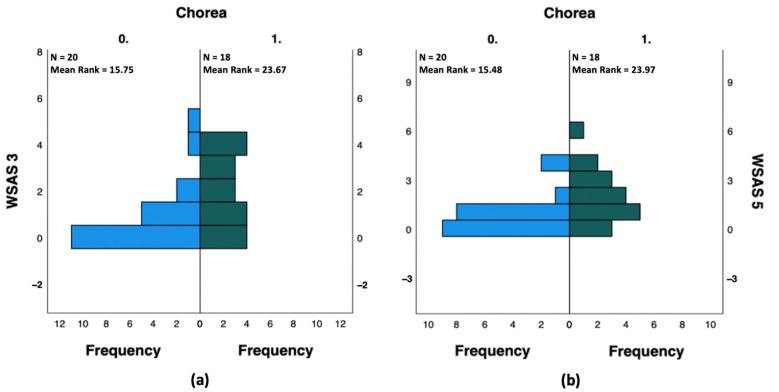
Independent-Samples Mann-Whitney U Tests reporting that (**a**) WSAS question 3 (private leisure) and (**b**) WSAS question 5 (close relationships) scores were significantly higher in SC patients (Chorea 1) were higher than those of nSC patients (Chorea 0).

**Table 1 ijerph-19-10586-t001:** Results of the chi-square test of independence showing that SC patients were significantly more likely than nSC patients to have neurologic symptoms other than chorea.

Chi-Square Tests					
	Value	df	Asymptotic Significance (2-Sided)	Exact Significance (2-Sided)	Exact Significance (1-Sided)
Pearson Chi-Square	19.368 ^a^	1	<0.001		
Continuity Correction ^b^	16.653	1	<0.001		
Likelihood Ratio	21.485	1	<0.001		
Fisher’s Exact Test				<0.001	<0.001
Linear-by-Linear Association	18.964	1	<0.001		
Number of Valid Cases	48				

^a^ 0 cells (0.0%) have expected count less than 5. The minimum expected count is 6.13. ^b^ Computed only for a 2 × 2 table.

**Table 2 ijerph-19-10586-t002:** Results of the chi-square test of independence showing that SC patients were significantly more likely than nSC patients to have neuropsychiatric symptoms.

Chi-Square Tests					
	Value	df	Asymptotic Significance (2-Sided)	Exact Significance (2-Sided)	Exact Significance (1-Sided)
Pearson Chi-Square	8.694 ^a^	1	0.003		
Continuity Correction ^b^	7.061	1	0.008		
Likelihood Ratio	9.034	1	0.003		
Fisher’s Exact Test				0.004	0.004
Linear-by-Linear Association	8.513	1	0.004		
Number of Valid Cases	48				

^a^ 0 cells (0.0%) have expected count less than 5. The minimum expected count is 6.13. ^b^ Computed only for a 2 × 2 table.

**Table 3 ijerph-19-10586-t003:** Descriptive study cohort table, showing the presence of several investigated variables in the overall population (n = 48), in SC patients (n = 21) and in nSC patients (n = 27). Significance calculated by the execution of chi-square tests of independence is also shown.

	Overall Population	SC	nSC	Significance (*p*)
Male gender	52.1%	33.3%	66.7%	0.045
Cardiac comorbidities	75.0%	57.1%	88.9%	0.029
Neurological symptoms	29.2%	61.9%	3.7%	<0.001
Neuropsychiatric symptoms	52.1%	76.2%	33.3%	0.008
OCD symptoms	4.2%	4.8%	3.7%	1.000
Mood or anxiety symptoms	41.7%	57.1%	29.6%	0.105
Psychotic symptoms	4.2%	9.5%	0.0%	0.186
Tics	16.7%	28.6%	7.4%	0.115
ADHD symptoms	4.2%	4.8%	3.7%	1.000
Conduct disorder symptoms	12.5%	23.8%	3.7%	0.073
Psychological counseling	37.5%	52.4%	25.9%	0.115
Neuropsychiatric tests	27.7%	50.0%	11.1%	0.009
Treatment	64.6%	85.7%	48.1%	0.017
Antiepileptic drugs	10.4%	23.8%	0.0%	0.012
Antipsychotic drugs	22.9%	52.4%	0.0%	<0.001
IVIG	8.3%	19.0%	0.0%	0.031
TALS-SR positivity	26.9%	45.5%	13.3%	0.095
PTSD A	19.2%	9.1%	26.7%	0.356
PTSD B	11.5%	9.1%	13.3%	1.000
Partial PTSD	30.8%	18.2%	40.0%	0.395
PTSD or partial PTSD	57.7%	63.6%	53.3%	0.701

## Data Availability

The data presented in this study are available within the article and the Appendix A.

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
