# Peer review of "Psychopathological Impact in Patients with History of Rheumatic Fever with or without Sydenham’s Chorea: A Multicenter Prospective Study"

_ijerph, 2022, doi:10.3390/ijerph191710586_

Round 1
Reviewer 1 Report
Orsini et al. presented a Italian multicenter prospective study on 48 patients with a previous diagnosis of acute rheumatic fever (ARF), with and without Sydenham’s chorea (SC), in order to assess the presence of neuropsychiatric symptoms after years of the onset of SC. The authors used eight different neuropsychological tests in each group (with and without SC).
They demonstrated the presence of neuropsychiatric symptoms in the SC population with statistically significant results.
The work was conducted with good methodology and has adequate references. A significant population was studied and the psychopathological impact of SC was demonstrated even after years of its onset.
Tips:
-Line 450: Insert references
-Insert a short table with the neuropsychiatric symptoms detected (rf: lines 195-205)
